# Carbene Addition Isomers of C_70_ formed in the Flame of Low-Pressure Combustion

**DOI:** 10.3390/nano12183087

**Published:** 2022-09-06

**Authors:** Fang-Fang Xie, Zuo-Chang Chen, You-Hui Wu, Han-Rui Tian, Shun-Liu Deng, Su-Yuan Xie, Lan-Sun Zheng

**Affiliations:** State Key Lab for Physical Chemistry of Solid Surfaces, Collaborative Innovation Center of Chemistry for Energy Materials, Department of Chemistry, College of Chemistry and Chemical Engineering, Xiamen University, Xiamen 361005, China

**Keywords:** low-pressure combustion, fullerene derivatives, in situ capture, carbene addition

## Abstract

In the flames during low-pressure combustion, not only a rich variety of fullerenes but also many reactive intermediates can be produced (e.g., carbene, CH_2_) that are short-lived and cannot be stabilized directly under normal circumstances. These intermediates can be captured by fullerene carbon cages for stabilization. In this paper, three C_71_H_2_ isomers were synthesized in situ in low-pressure benzene-acetylene-oxygen diffusion flame combustion. The results, which were unambiguously characterized by single-crystal X-ray diffraction, show that the three isomers are carbene addition products of *D_5h_*-C_70_ on different sites. The relative energies and stability of different C_71_H_2_ isomers are revealed by Ultraviolet-Visible (UV-Vis) absorption spectroscopy, in combination with theoretical calculations, in this work. Both the in situ capture and theoretical study of these C_71_H_2_ isomers in low-pressure combustion will provide more information regarding carbene additions to other fullerenes or other carbon clusters at high temperatures.

## 1. Introduction

The soot produced from the combustion of gaseous fuels at low pressure contains a rich variety of polycyclic aromatic hydrocarbon and fullerenes, including C_60_, C_70_, and other conventional IPR-satisfying (IPR—isolated pentagon rule) [1] fullerenes [2,3] and some special IPR-violating fullerenes [4,5,6,7]. Similar to the free-radical reactions in organic chemistry [8], electron-deficient fullerenes, such as C_60_ and C_70_, can act as free radical scavengers [9,10] to capture and stabilize the reactive species formed in the combustion process. The well-defined structures of fullerene derivatives provide important experimental evidence for the study of the combustion mechanism. C_60_(C_5_H_6_) was the first C_60_ derivative isolated from combustion products [11]. Our group reported a series of C_60_ derivatives involving methyl-substituted cyclopentadienyl (C_p_) species in a low-pressure acetylene-cyclopentadiene-oxygen flame, in the form of C_60_(C_p_)(CH_3_)*_n_* (*n* = 0–4), in which the proposed C1/C2 mechanism clearly explained the formation of curvatures in nanocarbon materials [3]. Although fullerene derivatives are prevalent, the chemical properties of C_70_ derivatives in low-pressure combustion flames have rarely been discussed. C_70_(C_14_H_10_) and C_70_(C_5_H_6_) [2] were isolated from the products of benzene-acetylene flame combustion, and the structures of these two C_70_ derivatives were determined in combination with theoretical simulation, i.e., the adducts obtained by the Diels–Alder reaction of the C_14_H_10_ and C_5_H_6_ groups with *D_5h_*-C_70_, respectively. However, the structures of other C_70_ derivatives in the flame are still unknown, mainly because the combustion products are very complex, making isolation and purification difficult.

*D_5h_*-C_70_ is the second most abundant product (after *I_h_*-C_60_) of the combustion products [12,13,14,15,16,17], with five unequal carbon atoms being present in its carbon cage, resulting in eight different C–C bonds (as shown in Figure 1). There are many reactive carbene intermediates seen in the process of low-pressure combustion, which can easily undergo addition reactions with the C–C bonds on the C_70_ cage to form ring-like exohedral derivatives. However, the carbene addition reactions of C_70_ involve chemo- and regioselectivity. The highest curvature in the region near the poles of the olive-shaped C_70_ brings unfavorable local strain, making it the most active chemical reaction site. Conversely, the equatorial region of C_70_ shows little local strain due to its having almost no curvature. The reactivity of the equatorial region is low [18,19,20,21,22,23,24] because the high activation energy barriers must be overcome for the reactions to occur.

Previously, Wang et al. reported the theoretical calculations of the carbene addition products of *D_5h_*-C_70_ [25], and found that eight possible C_71_H_2_ isomers could be formed by the addition reactions of carbenes with the eight different C–C bonds on C_70_, which are of types **a****–a**, **a****–b**, **b****–c**, **c****–c**, **c****–d**, **d****–d**, **d****–e,** and **e****–e**. Comparisons of the distance of the C-C bond at carbene insertion sites and the corresponding stabilization energies revealed that the isomers of **a****–a**, **b****–c**, **c****–d**, **d****–d,** and **e****–e** are opened structures of carbene addition, called homofullerenes, while the **a****–b**, **c****–c,** and **d****–e** isomers are conventional cycloaddition structures, known as methanofullerenes [26]. It has been reported that carbenes were added to the distinct sites of C_70_ to form various C_71_H_2_ isomers, which were subsequently fully characterized [18,19,20,21]. The synthesis of C_71_H_2_ could be achieved by solution chemical reactions [19,20,21], photochemical reactions [27], the Krätschmer–Huffman method [28,29], and pyrogenic synthetic methods [25]. However, the formation of C_71_H_2_ through the in situ capture of carbene by C_70_ during combustion has not yet been reported. In this paper, three C_71_H_2_ isomers (C_71_H_2_-I, C_71_H_2_-II, and C_71_H_2_-III) were successfully synthesized from low-pressure benzene-acetylene-oxygen diffusion flame combustion; these were definitely characterized as **c****–c**, **a****–b** and **e****–e** isomers by mass spectrometry, UV-Vis spectrometry, and single-crystal X-ray diffraction. The relative concentrations and stability of the three isomers were verified by theoretical calculations. The capture of these three carbene derivatives of *D_5h_*-C_70_ will be helpful for carbene addition studies at high temperatures.

## 2. Materials and Methods

### 2.1. Materials

All gases and other chemicals were purchased from commercial suppliers and were used as received. Benzene and toluene were bought from Sinopharm (Shanghai, China). Acetylene and oxygen were bought from Yidong Gas Co. (Xiamen, China). Toluene was used after distillation via standard procedures.

### 2.2. Synthesis and Separation

Carbon soot containing C_71_H_2_-I, C_71_H_2_-II, and C_71_H_2_-III was produced in the low-pressure diffusion flame combustion of benzene-acetylene-oxygen, using combustion equipment designed and built by our group [6]. During the synthesis process, the pressure inside the combustion chamber was controlled at about 25 Torr and the gas flow rates of vapored benzene, C_2_H_2_, and O_2_, were set to 1.0 L/min, 0.55 L/min, and 1.10 L/min, respectively. At least 500 g of soot was synthesized continuously at a yield of 3 g/h for the next separation.

The synthesized carbon soot was ultrasonically extracted 3–5 times, with toluene as solvent, and was then filtered and concentrated at room temperature. The toluene extracts were separated and purified, then analyzed with a Shimadzu LC-6AD high-performance liquid chromatograph (HPLC) (Shimadzu Co., Kyoto, Japan) in which toluene was used as the mobile phase; the Cosmosil Buckyprep column (i.d. 20 × 250 mm and i.d. 10 × 250 mm) and Cosmosil 5PBB column (i.d. 10 × 250 mm) were used alternately as stationary phases. The chromatogram was monitored at 330 nm during the separation. The procedures for product isolation are detailed in Appendix A, where mass spectrometry (MS) was used after each step to confirm the purity of the samples, ultimately obtaining C_71_H_2_-I, C_71_H_2_-II, and C_71_H_2_-III with high purity (up to 99%). The yields of C_71_H_2_-I, C_71_H_2_-II, and C_71_H_2_-III in carbon soot were roughly 0.035 mg, 0.18 mg, and 0.85 mg, respectively, which were considerably lower than those of C_60_ and C_70_ in the carbon soot (2.5 g and 1.25 g, respectively) (see Appendix A).

### 2.3. Characterization

Mass spectrometry (MS) was performed on a Bruker HCT mass spectrometer (Bruker Co., Karlsruhe, Germany), interfaced with an atmospheric pressure chemical ionization ion source (i.e., APCI-MS) in negative ion mode. Single-crystal X-ray diffraction data were collected on an Agilent SuperNova diffractometer (Agilent Technologies Ltd., Cheadle, UK) using a Cu Kα (λ = 1.54184 Å) microfocus X-ray source. The crystal data processing was carried out through CrysAlis^Pro3^. Within Olex2 [30], the structures were solved with the SHELXT and SHELXL-2015 [31] programs (George M. Sheldrick, Georg-August Universität Göttingen, Tammannstraße 4, Göttingen 37077, Germany), using the intrinsic phasing method, and refined using the full-matrix least-squares analysis based on F^2^. The SQUEEZE program, part of the crystallographic software PLATON package (Version: 91117, (C) 1980–2021 A.L.Spek, Utrecht University, Utrecht, The Netherlands) [32], was used to calculate the disordered area of solvent and remove its contributions from the intensity data, as needed. The UV-Vis spectra were recorded on a Shimadzu UV-2550 UV-Vis spectrophotometer (Shimadzu Co., Kyoto, Japan), with redistilled toluene as a solvent.

## 3. Results and Discussion

### 3.1. Mass Spectrometric Analysis of C_71_H_2_-I, C_71_H_2_-II, and C_71_H_2_-III

The molecular weights of C_71_H_2_-I, C_71_H_2_-II, and C_71_H_2_-III were identified by the negative ion mode of APCI-MS. The ionization temperature during the experiment was set to 250 °C. As shown in Appendix A, the three isomers of C_71_H_2_ had high purity; the molecular ion peaks of C_71_H_2_-I, C_71_H_2_-II, and C_71_H_2_-III are all around m/z 854.0, matching well with their molecular formula (C_71_H_2_). The insets in Appendix A demonstrate that the experimental mass spectra of the three isomers are consistent with the calculated peaks for C_71_H_2_.

The three isomers were also characterized by multi-stage mass spectrometry (MS/MS). As shown in Appendix A, no fragment ion peaks were found, indicating that they have good stability under high-energy conditions.

### 3.2. Crystallographic Identification of C_71_H_2_-I, C_71_H_2_-II, and C_71_H_2_-III

The black co-crystals of 2DPC{C_71_H_2_-I}, 2DPC{C_71_H_2_-II}, and 2DPC{C_71_H_2_-III} were formed by slow solvent evaporation from the mixed solutions of decapyrrylcorannulene (DPC) [33] and C_71_H_2_-I, C_71_H_2_-II, and C_71_H_2_-III in toluene, respectively. The structures of 2DPC{C_71_H_2_-I}, 2DPC{C_71_H_2_-II}, and 2DPC{C_71_H_2_-III} were unequivocally revealed by single-crystal X-ray diffraction (see Appendix A and Appendix A for the crystallographic details). Interestingly, the eutectic structures of these three isomers of C_71_H_2_ with DPCs indicate that the DPCs have very high flexibility. Palm-like DPCs can not only hold C_71_H_2_-I/III in a V-shape but also combine with C_71_H_2_-II in a parallel manner by adjusting the interactions between DPCs and C_71_H_2_ for high-quality eutectics. All DPCs and toluene molecules are omitted from Figure 2 for clarity.

It can be seen from Figure 2 that the pristine cages of C_71_H_2_-I, C_71_H_2_-II, and C_71_H_2_-III are IPR-satisfying *D_5h_*-C_70_. Due to the different addition sites of carbenes on the C_70_ cages, they are isomers of each other. The structures of C_71_H_2_-I, C_71_H_2_-II, and C_71_H_2_-III correspond to the **c****–c**, **a****–b,** and **e****–e** types of the C_71_H_2_ isomers, respectively, as demonstrated in the theoretical calculations [25]. Among them, the carbene additions of **c****–c** and **a****–b** isomers are similar to the “2 + 1” cycloaddition reactions in organic synthesis. Both **c****–c** and **a****–b** isomers exhibit *C_s_* symmetry and belong to the methanofullerenes. However, the carbene of the **e****–e** isomer is added at the equatorial site of C_70_ in an opened structure, resulting in the *C_2v_* symmetric homofullerene.

### 3.3. UV-Vis Spectra of C_71_H_2_-I, C_71_H_2_-II, and C_71_H_2_-III

The purified C_71_H_2_-I, C_71_H_2_-II. And C_71_H_2_-III could be dissolved in common solvents, such as toluene, benzene, *o*-dichlorobenzene, and carbon disulfide. The solutions of C_71_H_2_-I and C_71_H_2_-II are yellowish-brown, while the solution of C_71_H_2_-III is reddish-brown, similar to that of C_70_. As shown in Figure 3, the three isomers of C_71_H_2_ have obvious absorption peaks in the UV-visible region. It can be seen that the UV-Vis spectra of C_71_H_2_-I (**c****–c**) and C_71_H_2_-II (**a****–b**) are very similar, while the UV-Vis spectrum of C_71_H_2_-III (**e****–e**) is similar to that of *D_5h_*-C_70_. This phenomenon is consistent with Smith et al.’s previous proposal [20] that homofullerenes hold the π electron conjugation of the C_70_ skeleton to the greatest extent. In addition, the onset wavelengths of C_71_H_2_-I (**c****–c**), C_71_H_2_-II (**a****–b**), and C_71_H_2_-III (**e****–e**) gradually decrease, indicating that their optical energy gaps (Eg) increase progressively.

### 3.4. Theoretical Calculation Analysis

All possible C_71_H_2_ isomers have been optimized by the Gaussian 16 program (Version: A.03, Gaussian, Inc., Wallingford, CT, USA) at the B3LYP/6-31G(d,p) level. The orbital information and relative energies of them and of *D_5h_*-C_70_ are shown in Table 1 and Appendix A. Compared to *D_5h_*-C_70_, the stability of the carbene insertions slightly decreases because of the higher HOMO/LUMO energies. Of all the eight C_71_H_2_ isomers, **e****–e** is the most stable, while **d****–e** and **c****–d** are the least stable. With the insertion of carbenes, the distance of the C–C bonds at the insertion sites on *D_5h_*-C_70_ will change, where the distance of **e****–e** becomes 2.315 Å, while those of **a****–b** and **c****–c** are only 1.647 Å and 1.612 Å, respectively. The electrostatic potential surface details of the C_71_H_2_ isomers and *D_5h_*-C_70_ are shown in Appendix A, indicating that the electrostatic potentials of the carbene fragments are more positive than those of the rest of the caged fragments.

To gain a deeper insight into the relative abundances of all possible isomers of C_71_H_2_ at high temperatures, the rigid rotor and harmonic oscillator (RRHO) approximation [34] was employed and the Gibbs energies of isomers from 0 K to 5000 K were calculated. As shown in Figure 4, the **e–e** isomer is the most abundant among the C_71_H_2_ isomers over the entire temperature range, from 0 to 5000 K. As the temperature rises to around 3200 K, the relative concentrations of the other seven isomers tend to change to flat, at which point the **a–b** and **c–c** isomers become the second and third stable isomers after the **e****–e** isomer. With the temperature gradually increasing, the relative concentrations of **a****–a**, **c****–c,** and **d****–d** isomers change slightly but very closely, which corresponds to the relative energies resulting from the theoretical calculations in Table 1.

It can be seen that the capture of C_71_H_2_-I (**c****–c**), C_71_H_2_-II (**a****–b**), and C_71_H_2_-III (**e****–e**) is not accidental. According to the calculation results in Table 1 and Figure 4, **e****–e** (0.00 kcal/mol) and **a****–b** (9.88 kcal/mol) are the first and second stable isomers with the highest concentrations at high temperatures, respectively. Although the relative energies and relative concentrations of the **c****–c** and **d****–d** isomers are close to each other, the isomer captured in this work is **c****–c**. The reason for this may be that the C–C bond in *D_5h_*-C_70_ is an independent double bond (1.389 Å) at the **c****–c** site, while the **d****–d** site (1.434 Å) is only a small fragment of the aromatic region and carbenes prefer double bonds to the larger aromatic regions (Appendix A). Therefore, the **c–c** isomer may be more easily generated than the **d****–d** isomer under identical experimental conditions. Accordingly, the stability of C_71_H_2_-I (**c****–c**), C_71_H_2_-II (**a****–b**), and C_71_H_2_-III (**e****–e**) gradually increase. This was also verified by the UV-Vis spectral analysis in Figure 3, where the optical energy gaps (Eg) of C_71_H_2_-I, C_71_H_2_-II, and C_71_H_2_-III are presumed to increase gradually. Their ^1^H NMR spectra are also simulated by theoretical calculations (Appendix A). The other five isomers have not been isolated under the currently available synthetic conditions, probably because of their higher energies and lower stability than the three isomers obtained in this paper.

In the combustion processes of gaseous fuels, fullerenes might be produced from the continuous growth of intermediates, such as polycyclic aromatic hydrocarbons, bowl-shaped polycyclic aromatic hydrocarbons, and small carbon radical molecules [3,35,36,37,38,39,40]. The present work not only provides explicit structural characterizations of C_70_ carbene addition isomers but will also inspire further explorations into the capture of active intermediates and our understanding of the formation mechanism of fullerenes in combustion. Further investigations on the formation mechanism of fullerenes in combustion are still ongoing.

## 4. Conclusions

In summary, three C_70_ derivatives of carbene addition, C_71_H_2_-I (**c****–c**), C_71_H_2_-II (**a****–b**), and C_71_H_2_-III (**e****–e**), were successfully synthesized by low-pressure benzene-acetylene-oxygen diffusion flame combustion. The crystal structures of the three isomers have been unambiguously characterized by single-crystal X-ray diffraction. The crystallographic data show that C_71_H_2_-I and C_71_H_2_-II are methanofullerenes, while C_71_H_2_-III is a homofullerene. The band gaps and the relative stability that was estimated from the UV-Vis absorption spectra of the three isomers are in good agreement with the theoretical calculation results, indicating that their formation is not accidental. The synthesis of these three C_71_H_2_ isomers also indicated that the reactive and short-lived carbene intermediates in the flame of low-pressure combustion could be captured and stabilized by fullerene carbon cages. In the low-pressure flame from gaseous fuels, carbene and other small carbon radical molecules play very important roles in the growth of fullerenes. The current work could inspire further exploration of the capture of reactive intermediates and insights into the mechanism of fullerene formation in combustion.

## Figures and Tables

**Figure 1 nanomaterials-12-03087-f001:**
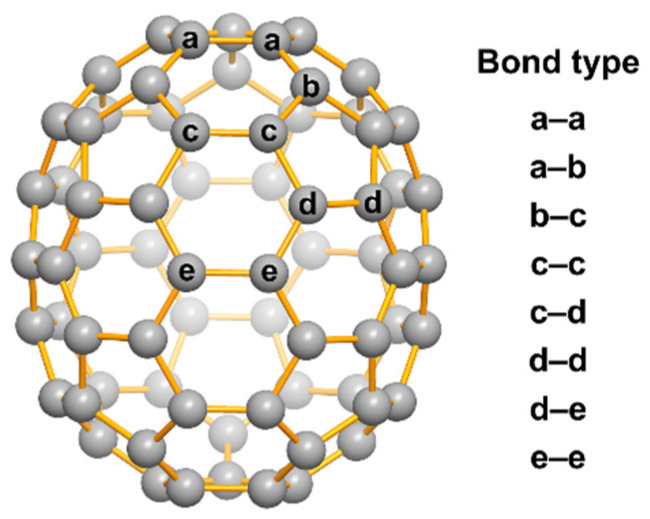
The five unequal types of carbon atoms (**a**, **b**, **c**, **d**, and **e**) and eight nonequivalent types of C–C bonds in C_70_.

**Figure 2 nanomaterials-12-03087-f002:**
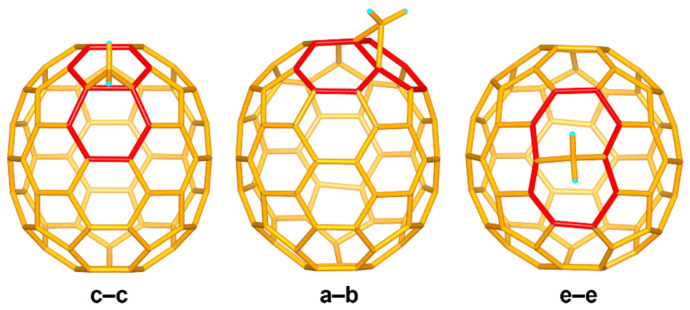
Ball-and-stick models of C_71_H_2_-I (**c****–c**), C_71_H_2_-II (**a****–b**), and C_71_H_2_-III (**e****–e**). C_71_H_2_-I and C_71_H_2_-II are methanofullerenes, C_71_H_2_-III is a homofullerene. The carbene addition sites are highlighted in red.

**Figure 3 nanomaterials-12-03087-f003:**
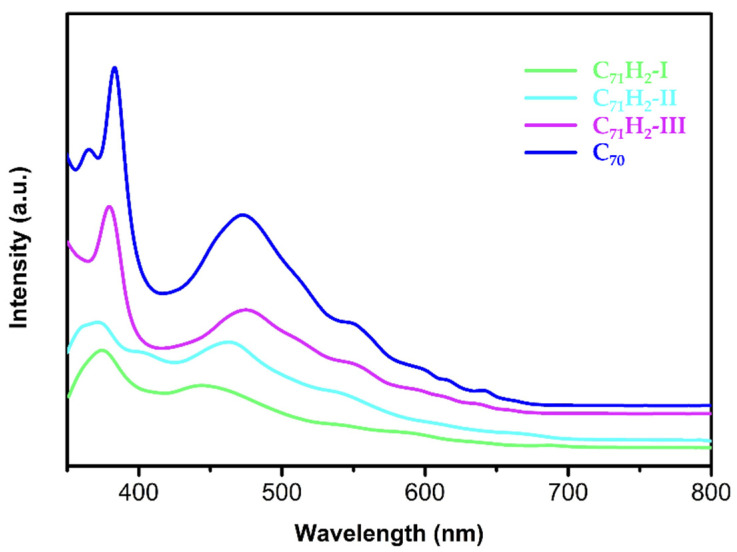
UV-Vis spectra of C_71_H_2_-I (**c****–c**), C_71_H_2_-II (**a****–b**), C_71_H_2_-III (**e****–e**), and *D_5h_*-C_70_ in toluene.

**Figure 4 nanomaterials-12-03087-f004:**
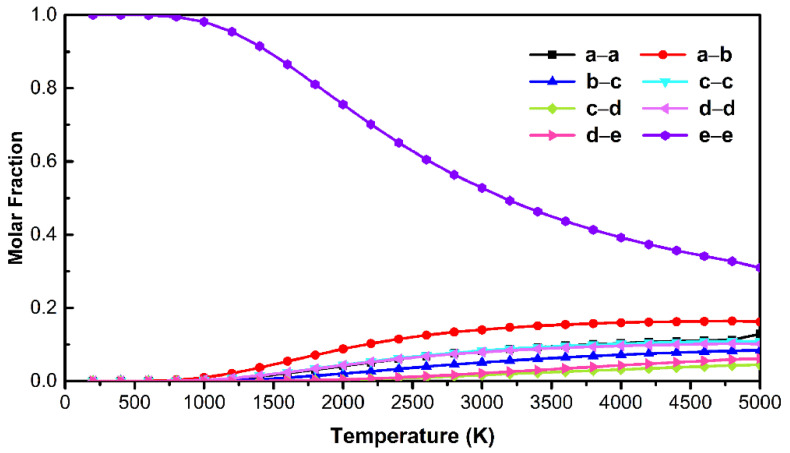
Relative concentrations for the C_71_H_2_ isomers, based on the RRHO approximation.

**Table 1 nanomaterials-12-03087-t001:** DFT calculation results of C_71_H_2_ isomers and *D_5h_*-C_70_ at the B3LYP/6-31G(d,p) level.

	HOMO	LUMO	H-L gap	R_C–C_ (Å)	∆*E* (kcal/mol)
*D_5h_*-C_70_	−5.92	−3.24	2.69	-	-
**e–e**	−5.83	−3.16	2.67	2.315 ^a^ (1.471 ^b^)	0.00
**a–b**	−5.66	−3.10	2.55	1.647 ^a^ (1.397 ^b^)	9.88
**d–d**	−5.72	−3.19	2.52	2.154 ^a^ (1.434 ^b^)	11.62
**c–c**	−5.64	−3.12	2.52	1.612 ^a^ (1.389 ^b^)	11.77
**a–a**	−5.78	−3.15	2.63	2.183 ^a^ (1.452 ^b^)	12.59
**b–c**	−5.78	−3.14	2.65	2.181 ^a^ (1.448 ^b^)	15.30
**c–d**	−5.71	−3.16	2.55	2.178 ^a^ (1.449 ^b^)	22.81
**d–e**	−5.74	−3.14	2.60	2.131 ^a^ (1.421 ^b^)	23.70

^a^ The C–C bond distance of the carbene insertion sites. ^b^ The relative C–C bond distance in pristine *D_5h_*-C_70_.

## Data Availability

The data presented in this study are available on reasonable request from the corresponding author.

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
