# Peer review of "Carbene Addition Isomers of C70 formed in the Flame of Low-Pressure Combustion"

_nanomaterials, 2022, doi:10.3390/nano12183087_

Round 1

Reviewer 1 Report

In this study by Xie and co-workers, the authors investigated the synthesis, separation, and characterization of the carbene addition isomers of C70 in the flame of low-pressure combustion.

The findings and discussion are an important information for fullerene chemistry. However, there are some unclear points listed below are supported to be explained before accepting the manuscript: 

1. Is it possible to provide quantitative yields for C60, C70, and three C71H2 isomers?

2. NMR is effective tools for structure analyses. The characterization of NMR spectra should be added in the results and discussion, and spectral data of C71H2 isomers should be added in the supporting information. 

3. C60 is the most abundant fullerenes, however, is there any reason why C61H2 is not detected? The information of C60 adducts was not shown. In addition, assignment of HPLC peaks, such as C60, C70, and these corresponding adducts, should be added in the figures.

4. The authors wrote that the capture of the three C71H2 isomers provides some experimental evidence for the study of the growth mechanism of fullerenes during low-pressure combustion. However, there is no sufficient explanation for the experimental results and growth mechanism.

5. The authors wrote that the demonstration plays a very important guiding role for the capture of other key intermediates. However, no information was provided on the effect of several synthetic conditions on the yield of C71H2.

6. If the addition configuration is 1,2-addition, it is better to label ‘d’ at the carbon pair adjacent to each other in Figure 1. 

7. The authors discussed the addition sites by relative energy of isomers, and bond length and size of aromatic region of C70. For these discission, the reviewer recommends the addition of the bond lengths and HOMO-LUMO of C70 in the manuscript or the supplementary materials file.

Reviewer 2 Report

This is a good manuscript on three isolated isomers of C71H2, formed by low-pressure combustion, with details of separation and Xray crystal structures.  The calculations at the B3LYP/6-31G(d,p) level are mostly a duplication of the previous calculations at the B3LYP/6-31G* level, but Fig. 4 is a very nice corollary, showing how the synthesis becomes less selective at higher temperature.

Round 2

Reviewer 1 Report

The authors have revised several parts to improve the manuscript. 

I recommend publication in nanomaterials after minor revision.

Some explanations toward reviewer comment are limited to the reviewer only. Some of them should be added in the manuscript to improve the manuscript.

The quantitative yields for each product are important information to understand these studies. In addition to the explanation written in the response to the reviewer’s comment 1, isolated yields (mg) of three C71H2 should be disclosed. Otherwise, it will cause misunderstanding for the reader, and the readers could not understand why NMR shows only theoretical results.

For comparison, the reviewer suggests adding unfunctionalized C70 data to Figures S6 and S7.
